# Potent Inhibitory Effect of BJ-3105, a 6-Alkoxypyridin-3-ol Derivative, on Murine Colitis Is Mediated by Activating AMPK and Inhibiting NOX

**DOI:** 10.3390/ijms21093145

**Published:** 2020-04-29

**Authors:** Pallavi Gurung, Sadan Dahal, Prakash Chaudhary, Diwakar Guragain, Ujjwala Karmacharya, Jung-Ae Kim, Byeong-Seon Jeong

**Affiliations:** College of Pharmacy, Yeungnam University, Gyeongsan 38541, Korea; gurungpallavi@ynu.ac.kr (P.G.); sadandahal@ynu.ac.kr (S.D.); prakash@ynu.ac.kr (P.C.); diwakarguragain@ynu.ac.kr (D.G.); ujjkarmacharya@hotmail.com (U.K.)

**Keywords:** AMP-activated protein kinase (AMPK), NADPH oxidase (NOX), colitis, colitis-associated tumor formation, tight junction molecule, BJ-3105

## Abstract

Inflammatory bowel disease (IBD) is a chronic relapsing inflammation in the gastrointestinal tract. Biological therapeutics and orally available small molecules like tofacitinib (a JAK inhibitor) have been developed to treat IBD, but half of the patients treated with these drugs fail to achieve sustained remission. In the present study, we compared the therapeutic effects of BJ-3105 (a 6-alkoxypyridin-3-ol derivative) and tofacitinib in IBD. BJ-3105 induced activation of AMP-activated protein kinase (AMPK) in the kinase activity measurement and recovery from cytokine-induced AMPK deactivation in HT-29 human colonic epithelial cells. Similar to tofacitinib and D942 (an AMPK activator), BJ-3105 inhibited IL-6-induced JAK2/STAT3 phosphorylation and TNF-α-stimulated activation of IKK/NF-κB, and consequently, stimulus-induced upregulations of inflammatory cytokines and inflammasome components. In addition, unlike tofacitinib or D942, BJ-3105 inhibited NADPH oxidase (NOX) activation and consequent superoxide production induced by activators (mevalonate and geranylgeranyl pyrophosphate) of the NOX cytosolic component Rac. In mice, oral administration with BJ-3105 ameliorated dextran sulfate sodium (DSS)-induced colitis and azoxymethane/DSS-induced colitis-associated tumor formation (CAT) much more potently than that with tofacitinib. Moreover, BJ-3105 suppressed the more severe form of colitis and CAT formation in mice with AMPK knocked-out in macrophages (*AMPKα*^fl/fl^-*Lyz2*-Cre mice) with much greater efficacy than tofacitinib. Taken together, our findings suggest BJ-3105, which exerted a much better anti-colitis effect than tofacitinib through AMPK activation and NOX inhibition, is a promising candidate for the treatment of IBD.

## 1. Introduction

The intestinal epithelium constitutes a frontline defense system against infectious and non-infectious threats by serving as an interface that enables crosstalk between the host immune system and the luminal environment of commensal microorganisms. Abnormalities in the functions and interactions between innate and adaptive immune systems in the gut epithelium lead to idiopathic chronic inflammation, which represents the pathogenesis of inflammatory bowel disease (IBD). IBD is a chronic relapsing inflammation of the gastrointestinal tract, and is divided based on lesion locations and histological features into two major disease types, namely, ulcerative colitis (UC) and Crohn’s disease (CD). Although specific triggers of immune system activation are largely unknown, IBD starts with activation of the innate immune system, which leads to epithelial inflammation followed by activation of the adaptive immune system through various feedback loops, and the generation of sustained chronic inflammation [1]. Resident intestinal macrophages located beneath the epithelium interact with epithelial cells, vascular cells, and other immune cells to generate immune-modulating effects and control tissue homeostasis [2,3]. In active IBD, macrophages derived from circulating monocytes are recruited to the mucosa and produce proinflammatory cytokines, chemokines, and reactive oxygen species (ROS). The secreted cytokines and chemokines, in turn, recruit more monocytes, neutrophils, and adaptive immune cells. Furthermore, continuous activation of inflammatory M1 macrophages is associated with tissue damage and IBD development, and ultimately with colitis-associated cancer [4,5]. 

Tumor necrosis factor (TNF)-α is an important proinflammatory cytokine that is chronically elevated in intestinal tissues and systemically in IBD patients [6,7]. TNF-α is mainly produced by macrophages, monocytes, T lymphocytes, and epithelial cells, and contributes to intestinal inflammation and damage by disrupting the intestinal barrier, inducing epithelial cell apoptosis, and upregulating adhesion molecules and other cytokines [8]. A previous report [9] showed that blockade of TNF-α activity improved intestinal barrier function by upregulating epithelial tight junction molecules and downregulating various inflammatory mediators and T cell activity. Since approval was granted for an anti-TNF-α antibody drug, anti-TNF-α therapy has been the first-line treatment for moderate to severe cases of IBD because it induces intestinal epithelium recovery and ameliorates clinical symptoms [10]. CD and UC are immunologically distinct IBD diseases. CD is primarily associated with T-helper type 1 (Th1) immune responses with elevated levels of IL-12, interferon (IFN)-γ, and TNF-α, whereas UC is described as a Th2-mediated disease in which IL-4, IL-5, and IL-13 levels are increased [11]. Although the underlying mechanism of the differential response is unclear, anti-TNF therapy is generally less effective in UC. In many IBD patients who do not respond to anti-TNF therapy, other inflammatory pathways, such as the interleukin (IL)-1β pathway, were upregulated [12]. 

In addition to TNF-α, other cytokines (e.g., IL-6 and IL-1β) also play a crucial role in the pathogenesis of IBD and mediate intracellular signaling by activating JAKs, a family of intracellular tyrosine kinases [13,14]. Tofacitinib is a JAK inhibitor approved for the treatment of rheumatoid and psoriatic arthritis, and has been shown to be effective in UC but not in CD [15]. Although tofacitinib is orally available and non-immunogenic, its limited efficacy demonstrates an unmet need for drugs that are more effective in IBD. Clinical data suggest a molecule that targets multiple signaling pathways might be a better proposition for IBD therapy. Such targets include AMP-activated protein kinase (AMPK), which inhibits TNF-α-stimulated IKK/NF-κB and IL-6-induced JAK/STAT3 signaling pathways [16]. AMPK also suppresses NADPH oxidase (NOX) activation [17] and inhibits redox-sensitive nuclear factor kappa B (NF-κB) [18,19,20], and thus inhibits the induction of NF-κB target genes, including TNF-α and IL-6 [21,22,23]. On the other hand, AMPK is inactivated by oxidative stress [24,25], and in line with the inverse relationship between AMPK and NOX, AMPK activation or NOX inhibition ameliorates dextran sulfate sodium (DSS)-induced colitis [26,27]. Thus, it appears a compound that activates AMPK and simultaneously inhibits NOX might be a promising therapeutic candidate for IBD.

In the present study, we report that BJ-3105 (a 6-alkoxypyridin-3-ol derivative), which has been previously shown to inhibit T cell differentiation and autoimmune encephalomyelitis by blocking cytokine-induced JAK phosphorylation [28], ameliorates DSS-induced colitis and azoxymethane/DSS-induced colitis-associated tumor formation in mice via the AMPK–NOX axis.

## 2. Results

### 2.1. BJ-3105 Blocked Inflammatory Cytokine-Induced Monocyte Adhesion to Colonic Epithelial Cells

AMPK is known to regulate multiple signaling pathways, including JAK/STAT, and in a previous study, BJ-3105 was found to inhibit the cytokine-induced JAK/STAT signaling pathway in T cells [28], suggesting the action of BJ-3105 is associated with AMPK activity. A cell-free kinase assay system was used to examine the ability of BJ-3105 to activate AMPK. As in metabolic stress conditions that lack ATP, AMPK is activated by an increasing concentration of AMP at a fixed concentration (25 μM) of ATP (Figure 1a). In the condition that only the fixed concentration of ATP without AMP was provided, the BJ-3105 concentration dependently (10–250 μM) increased AMPK activity (Figure 1b). D942 (an indirect AMPK activator) induced a slight increase in AMPK activity without showing concentration dependency, and this was similar to that observed for tofacitinib (Figure 1b). Next, BJ-3105 was evaluated for its inhibitory effect on TNF-α- and on IL-6-induced monocyte adhesion to HT-29 colonic epithelial cells, which is indicative of the early phase of colon inflammation and has been used to screen anti-IBD compounds [29,30]. Although BJ-3105 concentration dependently inhibited IL-6-induced monocyte-HT-29 cell adhesion, its inhibitory pattern differed from that of tofacitinib (Figure 1c). At concentrations lower than 1 μM, the inhibitory activity of tofacitinib was weaker than those of the other three drugs (AICAR, D942, and BJ-3105), whereas at concentrations > 1 μM, tofacitinib was a little bit better than the other inhibitors. In terms of TNF-α-induced adhesion of monocytes to HT-29 cells, the BJ-3105 concentration dependently inhibited adhesion at a level similar to those of tofacitinib, D942, and AICAR (Figure 1d). BJ-3105 was not toxic to CCD-841, a human normal colon epithelial cell line, and the IC_50_ concentration that inhibited cell viability by 50% was greater than 100 μM, similar to tofacitinib (Figure 1e).

### 2.2. Inhibitory Effects of BJ-3105 on the Expressions of Inflammatory Cytokines and Inflammasome Components 

Because the patterns of IL-6-induced cell adhesion by BJ-3105 and tofacitinib differed, we further compared their effects on IL-6-induced AMPK activity and gene expressions in HT-29 cells. IL-6 induced significant increases in the phosphorylations of JAK2 and STAT3 but significantly decreased AMPK activity. These changes were inhibited by BJ-3105, tofacitinib, and D942 (Figure 2a): BJ-3105 and tofacitinib were similarly effective and more effective than D942 (Figure 2b). In addition, BJ-3105 significantly blocked IL-6-induced upregulations of TNF-α, IL-6, and IL-10, and in this respect, it was more effective than the other two drugs. Next, we also examined the inhibitory effect of BJ-3105 on the formation of inflammasomes (the multiprotein complexes that activate caspase-1 and the maturation of IL-1β and IL-18). In HT-29 cells treated with *E. coli* strain BW25113, which mimics the condition of the colon mucosa, AMPK was inactivated and inflammasome components (NLRP3 and caspase-1), IL-1β, and IL-18 were upregulated (Figure 2c). BJ-3105 significantly inhibited the BW25113-induced changes with a much greater effect than tofacitinib (Figure 2d). 

In peritoneal macrophages treated with lipopolysaccharide (LPS; a well-known pathogen-associated entity expressed on Gram-negative bacteria), AMPK was deactivated, but this inhibition was recovered by BJ-3105 in a concentration-dependent manner (Figure 3a,b). Furthermore, LPS induced upregulations of both proinflammatory cytokines (TNF-α, IL-6, and IL-1β) and anti-inflammatory cytokines (IL-10 and TGF-β), and these cytokine upregulations were inhibited more potently by BJ-3105 than by tofacitinib (Figure 3b). 

Because the expressions of inflammatory cytokines (TNF-α and IL-6) and inflammasome-activated IL-1β and IL-18 are dependent on the activation of NF-κB [31], we compared the effects of BJ-3105, D942, and tofacitinib on TNF-α-induced NF-κB activation and AMPK inhibition in HT-29 cells. The recovery of AMPK activity from TNF-α-induced inhibition by BJ-3105 was similar to that of D942, but both were more effective than tofacitinib (Figure 4a,b). Similarly, the inhibitory effects of BJ-3105 on TNF-α inducing its own expression was greater than D942 or tofacitinib (Figure 4c). The TNF-α-induced increase in the phosphorylation of IKK (Figure 4d) and I-κB (Figure 4e) and decrease in I-κB protein level (Figure 4f) were also blocked by BJ-3105, D942, and tofacitinib, though BJ-3105 was more effective. Similarly, the TNF-α-induced nuclear translocation of NF-κB was significantly more inhibited by BJ-3105 than D942 or tofacitinib (Figure 4g). 

### 2.3. Inhibitory Effects of BJ-3105 on ROS Production and NOX Activity

NF-κB is activated by NOX-derived ROS in IBD [27,29,32], and the different patterns of activity shown by BJ-3105 on the one hand and D942 and tofacitinib on the other suggest BJ-3105 might regulate ROS production. In HT-29 cells treated with TNF-α for 3 h, ROS production (including superoxide and hydrogen peroxide) was inhibited by pretreating BJ-3105, D942, tofacitinib, and VAS2870 (a NOX2/4 inhibitor), and the inhibitory effect of BJ-3105 was greater than the other three drugs (Figure 5a,b). Because TNF-α-induced superoxide production is mainly mediated through NOX [29], we examined whether BJ-3105 and the other three drugs inhibit NOX-derived superoxide production. NOX is composed of membrane-bound subunits (p22phox and gp91phox) and cytosolic components (p47phox, p67phox, and p40phox), and its activation is initiated by the translocation of cytosolic components and Rac1 to the plasma membrane [33,34]. In the present study, we used mevalonate and geranylgeranyl pyrophosphate (GGPP), which activate NOX through the post-translational modification of Rac1, rather than by stimulating NOX via membrane receptors or cytosolic kinases [30,35]. Mevalonate-induced superoxide production was suppressed by BJ-3105 and VAS2870 but not by D942 and tofacitinib (Figure 5c). We also measured the concentration-dependent effects of BJ-3105, tofacitinib, and VAS2870 on NOX activation by using cell extracts that contained all NOX subunits, p22phox, gp9phox, p47phox, p67phox, p40phox, and Rac. GGPP-induced NOX activity was concentration dependently and similarly inhibited by BJ-3105 and VAS2870, whereas tofacitinib had no inhibitory effect (Figure 5d). NOX activation by TPA, which activates NOX through protein kinase C (PKC), was inhibited by BJ-3105, tofacitinib, and VAS2870, in a concentration-dependent manner, but BJ-3105 was more effective than VAS2870 and much more effective than tofacitinib (Figure 5e). 

### 2.4. BJ-3105 Ameliorated DSS-Induced Murine Colitis and Colitis-Associated Tumor Formation

We then investigated whether BJ-3105 exerts a better anti-colitis effect than tofacitinib, and the a more dependent effect on AMPK activation than NOX inhibition using DSS-treated colitis models in WT and AMPKαfl/fl-Lyz2-Cre mice (AMPK was specifically knocked-out in macrophages). DSS induced more severe colitis in AMPKαfl/fl-Lyz2-Cre mice, as assessed by body weight loss (Figure 6a), increases in colon weight/unit length (Figure 6b), and MPO levels, than in WT mice (Figure 6c). Oral administration with BJ-3105 dose dependently suppressed DSS-induced colitis in WT and AMPKαfl/fl-Lyz2-Cre mice. In line with the observed greater activity of BJ-3105 in terms of activation of AMPK in vitro than tofacitinib, DSS-induced deactivation of AMPK in colon tissues was recovered to more than the control level by BJ-3105 (10 mg/kg) and tofacitinib (30 mg/kg), though this recovery was greater for BJ-3105 (Figure 6d,e). In colon tissues, DSS-induced upregulations of NOX1 and NOX2 in AMPKαfl/fl-Lyz2-Cre mice and NOX2 in WT mice (Figure 6f) were suppressed by BJ-3105. In addition, DSS-induced downregulations of E-cadherin and claudin-2 (epithelial junction molecules) in colon tissues were significantly blocked by BJ-3105 and less potently by tofacitinib (Figure 6g). Moreover, the TNF-α levels of colon tissues in AMPKαfl/fl-Lyz2-Cre mice were much higher than in WT mice, and these increases were also significantly inhibited by BJ-3105 much greater than by tofacitinib. On the other hand, colonic IL-10 levels in WT mice were increased by DSS, and this increase was suppressed by BJ-3105 and tofacitinib. However, IL-10 levels in AMPKαfl/fl-Lyz2-Cre mice were barely detectable, and this was not changed by DSS or drug treatments (Figure 6i). Furthermore, DSS-induced degradation of I-κB (Figure 6d) and consequently activation and nuclear translocation of redox-sensitive transcription factor NF-κB were suppressed by BJ-3105 much more effectively than by tofacitinib (Figure 6j). 

We further investigated whether the anti-colitis efficacy of BJ-3105 extended to the inhibition of tumor formation caused by AOM pretreatment and chronic DSS treatment. At the end of the experiment after nine weeks of DSS and drug treatments, body weights were significantly decreased, and the decrease was more severe in AMPKαfl/fl-Lyz2-Cre mice than in WT mice (Figure 7a). In the colon tissues, many tumors were developed in WT and AMPKαfl/fl-Lyz2-Cre mice, but higher numbers of tumor were found in AMPKαfl/fl-Lyz2-Cre mice (Figure 7b). In addition to recovering body weights, BJ-3105 and tofacitinib also significantly suppressed AOM/DSS-induced tumor formation (Figure 7b). Furthermore, the recovery effect of BJ-3105 (3 mg/kg) was greater than that of tofacitinib (30 mg/kg), and much superior to that of sulfasalazine, a mild acting anti-colitis drug.

## 3. Discussion

The present study shows BJ-3105 (chemical name: 2,4,5-trimethyl-6-(3-phenylpropoxy)pyridin-3-ol) directly activates AMPK and simultaneously inhibits NOX activity, and inhibits the JAK/STAT pathway more effectively than tofacitinib. In addition, more severe colitis and colitis-associated colon tumor formation in AMPK-deleted mice than in WT mice suggested NOX upregulation was more associated with DSS-induced colitis than AMPK activity downregulation. Furthermore, the similar inhibitory effects of BJ-3105 on WT and AMPKαfl/fl-Lyz2-Cre mice treated with DSS demonstrated that NOX inhibition was primarily responsible for the efficacy of BJ-3105 against DSS-induced mouse colitis and colitis-associated colon tumor formation.

In IBD, various inflammatory cytokines are overexpressed by immune cells in inflamed tissues [36], and activate transcellular signaling pathways, including the JAK/STAT pathway. In the present study, colonic epithelial cells and macrophages were found to produce inflammatory cytokines, TNF-α, IL-6, IL-1β, IL-10, and TGF-β, in response to cytokine or antigenic stimulation, and this was also observed in the colon tissues of DSS-induced colitis, though proinflammatory TNF-α levels were much greater than anti-inflammatory IL-10 levels. Although IL-10 level changes in inflamed colon tissues depend on the colitis model [37,38], the present study shows IL-10 levels were increased by DSS in the colon tissues of WT mice. In addition, like proinflammatory cytokines (TNF-α and IL-6), IL-10 was also upregulated by IL-6 and by LPS in HT-29 human colonic epithelial cells. Suppressions of these cytokine expressions by tofacitinib and D-942 (an AMPK activator) indicated cytokine expressions, including that of IL-10, were regulated by JAK/STAT and AMPK signaling pathways. Consistent with previous findings that AMPK regulates the JAK-STAT pathway, a crucial driver of chronic inflammatory and malignant diseases [39,40], our current study also showed complete suppression of IL-10 expression in colon tissues of AMPK-deleted mice, which still express JAK2 and STAT3 proteins, indicating that without AMPK, IL-10 production is not possible. 

AMPK activity was recovered from TNF-α- or IL-6-induced inhibition by BJ-3105 and tofacitinib in HT-29 cells. However, the AMPK kinase activity assay showed BJ-3105 directly activated AMPK, whereas tofacitinib did so indirectly. The more potent effects of BJ-3105 on colitis and inflammatory cytokine expression than tofacitinib suggested BJ-3105 possessed an activity other than AMPK-activating activity. Notably, we observed that BJ-3105 inhibited NOX activation. Although BJ-3105 and tofacitinib inhibited intracellular ROS production by TNF-α, their inhibitory effects on superoxide production by an intracellular stimulator (mevalonate, GGPP, or TPA) were not the same. BJ-3105, like VAS2870, inhibited NOX activation induced by mevalonate, GGPP, and TPA, which have different modes of action. Because mevalonate and GGPP activate NOX by activating Rac, whereas TPA activates NOX via protein kinase C, the inhibitory effects of BJ-3105 on NOX activation indicate that the action of BJ-3105 may be mediated by direct inhibition of the NOX catalytic subunit or by inhibition of cytosolic component translocation to the cell membrane. In contrast, tofacitinib failed to inhibit mevalonate or GGPP-induced NOX activation but inhibited TPA-induced NOX activation. Tofacitinib is known to bind to the ATP binding site of the JAK kinase domain [41], which suggests the inhibitory effect of tofacitinib on NOX activation is mediated through the inhibition of PKC (an intracellular kinase). This differential inhibitory effect of BJ-3105 and tofacitinib on NOX activation corresponded to the difference in efficacy between the two drugs on inflammatory cytokine expression, colitis, and colitis-associated tumor formation in AMPK-deleted mice showing more enhanced NOX expression. 

Our findings suggest a negative correlation exists between inflammasome induction and AMPK activity. Intestinal epithelial cells express a variety of innate immune receptors, which sense pathogen-associated or damage-associated molecular patterns via pattern recognition receptors (PRRs), such as toll-like receptors (TLRs) [42]. Stimulated PRRs in intestinal epithelial cells activate inflammasome formation and trigger the expression and secretion of inflammatory cytokines and chemokines, which prime the immune cell response under the epithelium [43]. In addition, inflammasome activation by TLRs induces inflammatory cell death via the activation of caspase-1 and IL-1β [44]. These processes impair the barrier function of the intestinal epithelium and contribute to the development of diseases, including IBD [45]. During the TLR-activated process leading to intestinal barrier leakage, AMPK is deactivated. In the current study, LPS-induced AMPK deactivation and inflammasome component upregulations were suppressed by BJ-3105 more so than by tofacitinib. As TLR activation is associated with NOX activation and oxidative stress [46], the suppression of inflammasome-induced IL-1β expression by BJ-3105 indicates that epithelial dysfunction and inflammasome inductions were regulated by NOX-induced ROS and AMPK inhibition. Similarly, BJ-3105 restored colon epithelial junction molecules’ expression (E-cadherin and claudin-2) from the DSS-induced decreased expression. 

In conclusion, our current study demonstrated that inflammatory cytokine upregulation, inflammasome formation, and consequent colitis induction were under the control of the NOX-AMPK axis, and showed that BJ-3105, an AMPK activator and NOX inhibitor, was much more effective than tofacitinib at ameliorating colitis and colitis-associated tumor formation in mice.

## 4. Materials and Methods 

### 4.1. Materials

Chemical reagents were obtained from Sigma–Aldrich (St. Louis, MO, USA) unless otherwise specified. RPMI1640, fetal bovine serum (FBS), and penicillin/streptomycin were purchased from Invitrogen Life Technologies (Carlsbad, CA, USA). Trypsin/EDTA was purchased from GE Healthcare Life Sciences (Logan, UT, USA). E-Cadherin rabbit polyclonal antibody was purchased from Santa Cruz Biotechnology (Santa Cruz, CA, USA). AMPK-α rabbit polyclonal antibody, rabbit monoclonal antibodies specific to p-AMPK-α (Thr172), JAK2, p-JAK2 (Tyr1008), and occludin, and mouse monoclonal antibodies detecting STAT3, p-STAT3 (Tyr705), cleaved caspase 1, and IL-1β were purchased from Cell Signaling Technology Inc. (Boston, MA, USA). NOX2 and ICAM-1 rabbit monoclonal antibodies, and TGF-β, TNF-α, NOX1, p-p47phox, pro-caspase-1, and claudin 2 rabbit polyclonal antibodies were purchased from Abcam (Cambridge, MA, USA). Rabbit polyclonal antibodies specific for IL-10 or IL-6 were from Abbiotec (San Diego, CA, USA), and NLRP3 rabbit polyclonal antibody was purchased from Novus Biologicals (Centennial, CO, USA). Tofacitinib citrate salt was obtained from L.C Laboratories (Woburn, MA, USA). VAS2870 was purchased from Sigma-Aldrich (St. Louis, MO, USA). BJ-3105 was synthesized by Byeong-Seon Jeong as reported [47].

### 4.2. Cell Culture

HT-29 (a human colonic epithelial cell line), U937 (a human pre-monocytic cell line), and CCD-841 (a human normal colon epithelial cell line) were obtained from the American Type Culture Collection (Manassas, VA, USA). Cells were cultured in RPMI-1640 (HT-29 and U937) and DMEM high-glucose (CCD 841) media containing 10% FBS, 100 IU/mL of penicillin, and 100 μg/mL of streptomycin and a maintained atmosphere in 5% CO_2_ humidified at 37 °C. 

K-12 BW25113/pCM18 cells (*E. coli* cells) provided by Dr. Jintae Lee (Yeungnam University, Korea) were grown for 16 h at 37 °C in LB medium, harvested by centrifugation at 16,200 *g* for 5 min at 4 °C, washed once with phosphate-buffered saline (PBS), and resuspended in PBS. *E. coli* cells (1 × 10^8^) were added to wells containing a confluent monolayer of HT-29 cells and incubated for 3 h. HT-29 cells were then washed three times with PBS to remove non-adherent *E. coli*. 

Murine peritoneal macrophages were isolated as described previously [48] from 6- to 7-week-old female C57BL/6 mice. Briefly, 1 mL of 4% (*w*/*v*) Brewer thioglycolate medium (Difco, Detroit, MI, USA) was injected intraperitoneally to induce macrophage accumulation in the peritoneal cavity. Three day after injection, cells that accumulated in the peritoneal cavity were isolated by injecting and washing the cavity with 10 mL of calcium and magnesium free ice-cold Hank’s balanced salt solution using a 26G needle and syringe. Cells were collected using a 21G needle and centrifuged at 1500 rpm for 5 min, and the cell pellets so obtained were resuspended in red blood cell lysis buffer, and recentrifuged. Cell pellets were then resuspended in RPMI 1640 medium containing 10% FBS and 1% penicillin-streptomycin (P/S) and cells were seeded in 60-mm culture dishes at 1 × 10^5^ cells/cm^2^ and incubated at 37 °C in a 5% CO_2_ humidified atmosphere.

### 4.3. AMPK Activity Measurement 

AMPK kinase assays was performed using an AMPK (A1B1G1) kinase enzyme system (Promega, Madison, WI, USA) linked to an ADP-GlowTM kinase assay kit (Promega), according to the manufacturer’s instruction. Briefly, the enzymatic reaction was initiated by adding 25 μM ATP to a mixture consisting of 4 ng/μL AMPK, 0.2 μg/μL SAMStide (AMPK substrate included in the kit), and a test drug (AICAR, D942, VAS2870, tofacitinib, or BJ-3105). AMP (100 μM) was used as the positive control. Reactions were conducted for 60 min at 25 °C, and then, 25 μL of ADP-GlowTM reagent were added to the mixtures and incubated for 40 min at 25 °C. After incubation, 50 μL of kinase detection reagent were added and incubated for 30 min at 25 °C. Finally, luminescence was measured using a Fluostar Optima microplate reader (BMG LABTECH GmbH, Germany).

### 4.4. Monocyte–Epithelial Cell Adhesion Assay

Monocyte to colon epithelial cell adhesion assays were performed using U937 pre-monocytic human cells prelabeled with 2’,7’-bis (2-carboxyethyl)-5(6) carboxyl fluorescein acetoxymethyl ester (10 µg/mL) as previously reported [49,50] with a slight modification. HT-29 cells (2 × 10^5^ cells/well) cultured in 48-well plates were pretreated with compounds for 1 h. Prelabeled U937 cells were then seeded (5 × 10^5^ cells/well) onto HT-29 cell monolayers and treated with TNF-α (10 ng/mL) for 3 h at 37 °C. Non-adhering U937 cells were removed by washing three times with PBS. Cells were lysed with 0.1% Triton X-100 in Tris (0.1 M) in a shaker for 30 min at 25 °C. Fluorescence intensities were then measured using a Fluostar Optima microplate reader at excitation and emission wavelengths of 485 and 520 nm, respectively.

### 4.5. Cytotoxicity Measurement

CCD 841 cells were seeded in 96-well plates and treated with or without drugs in media containing 1% FBS for 48 h. Cell viability was measured using 3-(4,5-dimethylthiazol-2-yl)-2,5-diphenyl tetrazolium bromide (MTT) assay. After 4 h of forming formazan crystal by adding MTT, DMSO was added, and absorbance was measured at 540 nm using a Fluostar Optima microplate reader. 

### 4.6. Western Blotting

Cells scraped from culture dishes were centrifuged at 900× *g* for 5 min and pellets so obtained were dissolved in RIPA buffer containing protease and phosphatase inhibitors. Total protein concentrations were determined using BCA protein assay reagent (Pierce, Rockford, IL, USA). Proteins were separated by SDS-PAGE and transferred at 200 mA onto Hybond ECL nitrocellulose membranes (Amersham Life Science, Buckinghamshire, UK) for 1 h. Non-specific binding was blocked by using 5% bovine serum albumin (BSA) in Tris-buffered saline (TBS)-Tween 20 (TBS-T) for 1 h. Membranes were incubated with primary antibody in BSA or skim milk-TBS overnight at 4 °C, and washed three times with TBS-T prior to incubation with horseradish peroxidase-conjugated secondary antibody for 1 h at room temperature. Membranes were washed, incubated with ECL (Pierce) detection reagent, and exposed under a luminescent image analyzer, LAS-4000 mini (Fuji, Japan). β-Actin was used as the loading control.

### 4.7. Intracellular ROS Measurement

Intracellular ROS levels, including superoxide anion and hydrogen peroxide, were measured using dichlorofluorescein diacetate (DCF-DA, a fluorescent dye). Serum-starved HT-29 cells were treated with TNF-α in the presence or absence of compounds. After incubation for 3 h, DCF-DA (5 μM) was added at 37 °C, and cells were photographed using an inverted fluorescent microscope (TE2000-U; Nikon, Japan). Fluorescence intensity was quantitated in another set of cells prepared in the same manner using a FLUROstar Omega microplate reader.

Superoxide anions were measured as previously published with a slight modification [51]. Briefly, HT-29 cells (1 × 10^5^ cells/well) were seeded in white opaque 96-well plates. On the next day, cells were pretreated with drugs for 5 min prior to the treatment with stimulators, that is, mevalonate or geranylgeranyl pyrophosphate, for 5 min. Chemiluminescence was measured using lucigenin (400 μM) in a FLUROstar Omega microplate reader.

### 4.8. NOX Activity Measurement 

HT-29 cells were harvested using Krebs-HEPES buffer (pH 7.4) containing a protease and phosphatase inhibitor cocktail (Thermo Fisher Scientific, Waltham, MA, USA), homogenized with Dounce homogenizer, and centrifuged at 10,000× *g* for 15 min. Protein concentrations were measured using a BCA protein assay kit (Thermo Fisher Scientific). Equal amounts of protein extracts were transferred to wells of 96-well plates with white-coated bottoms. NADPH (100 μM) was added immediately after BJ-3105, tofacitinib, or VAS2870 treatment. TPA or GGPP were then added and incubated for 5 min at 37 °C. Chemiluminescence was measured using a FLUROstar Omega microplate reader (BMG Labtech GmbH, Offenburg, Germany).

### 4.9. Induction of Colitis and Colitis-Associated Tumor Formation in Mice

Age-matched female WT and AMPKαfl/fl-Lyz2-Cre mice with a C57BL/6 background were used in the study. Lyz2-Cre KO and AMPKαfl/fl murine lines were purchased from the Jackson Laboratory (Bar Harbor, ME, USA). AMPKαfl/fl-Lyz2-Cre mice were produced by crossing AMPKαfl/fl mice and Lyz2-Cre mice. Genotyping was performed by amplifying genomic DNA extracted from tail tissue (Table 1 for primer sequences). All mice were kept in a pathogen-free facility at the Animal Centre of Yeungnam University. 

DSS-induced colitis was induced in WT and AMPKαfl/fl-Lyz2-Cre mice. Control mice were given normal drinking water throughout the experiment, while mice in experimental groups were given 2% (*w*/*v*) DSS (MW 36–50 KDa; MP Biomedicals, Solon, OH, USA) solution in drinking water for 7 days. Drinking bottles were changed every third day. Body weights, stool consistencies, and the presence of fecal blood were recorded daily. Tofacitinib or BJ-3105 were orally administered daily starting 6 days after DSS treatment for 7 days. At the end of the experiments, mice were sacrificed by CO_2_ inhalation. Gross morphologies and myoperoxidase levels of colon tissues were determined. 

Thirty 6- to 8-week-old female mice were pooled and divided into 6 groups in WT and Lyz2cre-AMPKfl/fl mice for the colitis-associated cancer model. One group was the sham-operated normal control group (AOM-DSS-untreated). The other five groups were administered azoxymethane (AOM, a procarcinogen) at 10 mg/kg by a single intraperitoneal injection. A week later, the AOM-treated groups were treated with the first round of a 7-day DSS treatment (2% DSS in drinking water). The second and third rounds of DSS were administered at 4 and 7 weeks following AOM treatment using 2% DSS and 1% DSS, respectively. Tofacitinib (30 mg/kg), sulfasalazine (300 mg/kg), and low-dose (1 mg/kg), and high-dose BJ-3105 (30 mg/kg) were orally administered daily for 9 weeks (6 days per week) starting on the first day of DSS treatment. On the 71st day, mice were sacrificed, and colon tissues were excised and examined for tumor formation. 

All animal experiments were approved beforehand by the Institutional Animal Care and Use Committee of Yeungnam University and were performed accordingly the guidelines issued by the Institute of Laboratory Animal Resources (1996) and Yeungnam University (the care and use of animals (2009)).

### 4.10. Measurement of MPO 

MPO levels were measured in homogenates using the entire tissues of the colon wall. Briefly, colon tissue (50 mg) was mixed with 300 μL lysis buffer and homogenized using a Bead blaster 24 (Benchmark Scientific, NJ, USA). MPO levels in collected supernatants were determined using an MPO Assay Kit (Hycult Biotech, Uden, The Netherlands). 

### 4.11. Enzyme-Linked Immunosorbent Assay (ELISA) 

ELISA was performed to determine the levels of IL-10 and TNF-α in the homogenized colon tissues of DSS- and 5-HT-treated mice. Colon tissue (50 mg) was mixed with 300 μL of lysis buffer and homogenized by using Bead blaster 24. Collected supernatants were subjected to ELISA. The ELISA kits for mouse TNF-α and mouse IL-10 were obtained from R&D Systems (Minneapolis, MN, USA). Supernatant samples, standards, and controls were added to pre-coated ELISA plates, incubated at 25 °C for 2 h, washed five times, treated with mouse-compound (IL-10 and TNF-α) conjugate (Quantikine ELISA kit, R&D Systems, Minneapolis, MN, USA) for 2 h at 25 °C, rewashed, and treated with substrate solution for 30 min at room temperature. Stop solution was then added and optical densities were determined using a Fluostar Optima microplate reader. Concentrations were determined using standard curves generated with data from TNF-α (0–700 pg/mL) and IL-10 (0–1000 pg/mL).

### 4.12. Statistics 

Statistical significances between groups were determined using one-way or two-way ANOVA followed by the Student–Newman–Keul’s comparison method (GraphPad Prism 5.0 software, San Diego, CA, USA). Results are presented as the means ± standard errors of at least three independent experiments. Statistical significance was accepted for *p* values < 0.05. 

## Figures and Tables

**Figure 1 ijms-21-03145-f001:**
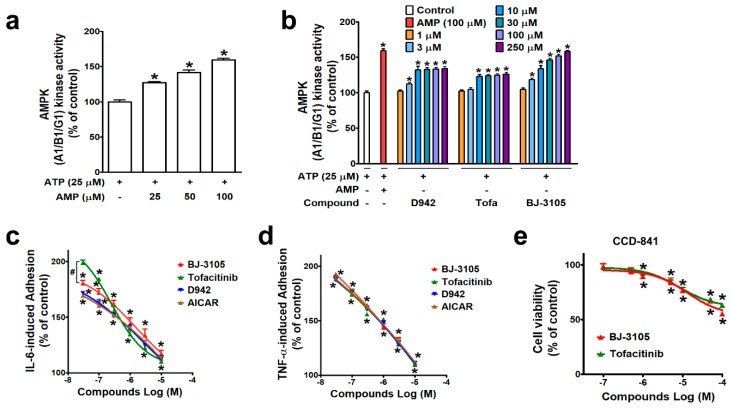
Comparison of the effects of BJ-3105 and tofacitinib on AMPK activity and cytokine-induced adhesion of monocytic cells to colonic epithelial cells. (**a**,**b**) AMPK enzyme activity at an ATP concentration of 25 μM was measured using an in vitro enzyme assay kit (AMPK A1B1G1 kinase enzyme system) in the presence of different concentrations of AMP (**a**) or various concentrations of D942, VAS2870, tofacitinib, and BJ-3105 (**b**). Results are presented as the means ± SEMs of at least three independent experiments performed in triplicate. **p* < 0.05, compared to the vehicle-treated control group. (**c**,**d**) Inhibitory effects of BJ-3105, tofacitinib, D-942, and AICAR on IL-6- (**c**) and on TNF-α-induced (**d**) U937 cell adhesion to HT-29 cells. BJ-3105, tofacitinib, D-942, and AICAR were pretreated for 1 h, and treated with TNF-α or IL-6 for 3 h. Results are presented as the means ± SEMs of at least three independent experiments. **p* < 0.05, versus the vehicle-treated control group. ^#^*p* < 0.05, versus the tofacitinib- or D942-treated group. (**e**) Cytotoxic effect of BJ-3105 and tofacitinib in CCD-841, a normal epithelial colon cell line. Cells were treated with BJ-3105 or tofacitinib for 48 h. **p* < 0.05, versus the vehicle-treated control group.

**Figure 2 ijms-21-03145-f002:**
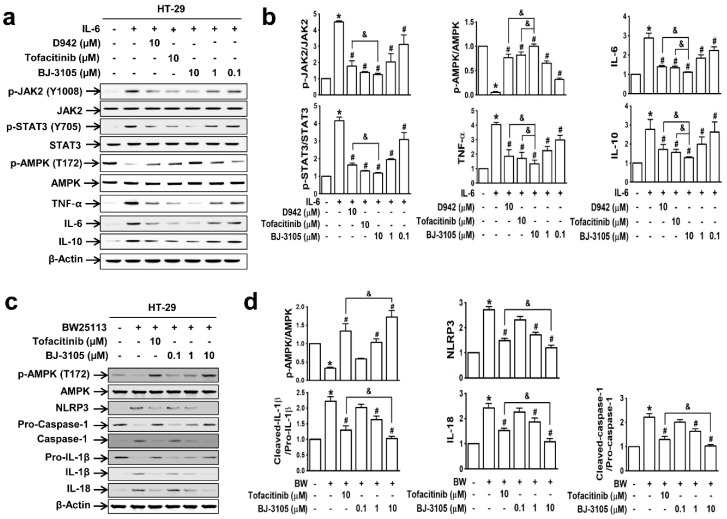
BJ-3105 blocked IL-6- or *E. coli* BW25113-induced AMPK inhibition and upregulations of cytokines and inflammasome better than tofacitinib in HT-29 cells. (**a**,**b**) Immunoblots (**a**) and quantitation (**b**) of IL-6-induced phosphorylation of JAK, STAT, and AMPK, and expressions of inflammatory cytokines. * *p* < 0.05, versus the vehicle-treated control group. ^#^*p* < 0.05, versus the IL-6-treated group. ^&^*p* < 0.05, versus the tofacitinib-treated group. (**c**,**d**) HT-29 cells were prereated with BJ-3105 or tofacitinib for 1 h prior to commensal bacteria (*E. coli* strain BW25113) for 3 h. After HT-29 cells were washed three times with PBS to remove non-adhering *E. coli*, the cells were used for immunoblotting. Immunoblots (**c**) of AMPK and inflammasome components, and their quantitation (**d**). Results are presented as the means ± SEMs of at least three independent experiments. **p* < 0.05, versus the vehicle-treated control group. ^#^*p* < 0.05, versus the BW25113-treated group. ^&^*p* < 0.05, versus the tofacitinib-treated group.

**Figure 3 ijms-21-03145-f003:**
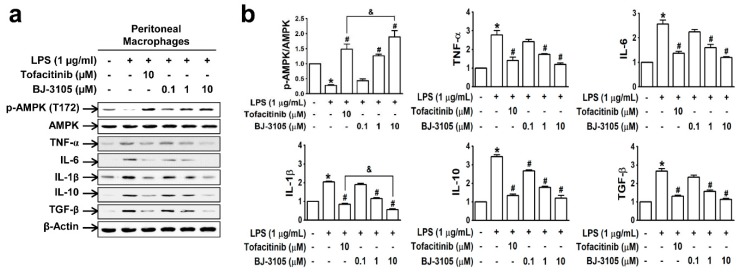
Effects of BJ-3105 and tofacitinib on LPS-induced AMPK activity and inflammatory cytokine expressions in peritoneal macrophages. (**a**) AMPK and inflammatory cytokine expression levels were analyzed by immunoblotting. (**b**) Bar graphs represent averaged quantitation of the immunoblots from at least three independent experiments. **p* < 0.05, versus the vehicle-treated control group. ^#^*p* < 0.05, versus the BW25113-treated group. ^&^*p* < 0.05, versus the tofacitinib-treated group.

**Figure 4 ijms-21-03145-f004:**
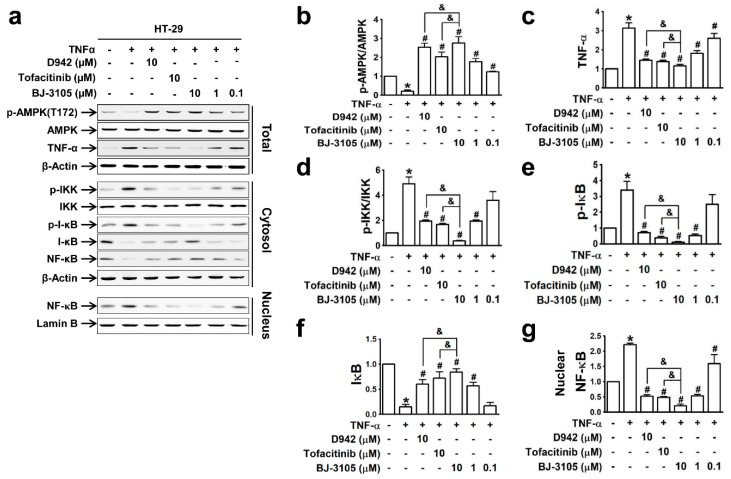
Inhibitory effects of BJ-3105, tofacitinib and D-942 on TNF-α-induced changes in the phosphorylation of AMPK, IKK, I-κB, and nuclear translocation of NF-κB in HT-29 cells. Immunoblots (**a**) and quantitation of expression levels of AMPK (**b**), TNF-α (**c**), p-IKK (**d**), p-I-κB (**e**), I-κB (**f**) and nuclear NF-κB (**g**). **p* < 0.05, versus the vehicle-treated control group. ^#^*p* < 0.05, versus the TNF-α-treated group. ^&^
*p* < 0.05, versus the tofacitinib-treated group.

**Figure 5 ijms-21-03145-f005:**
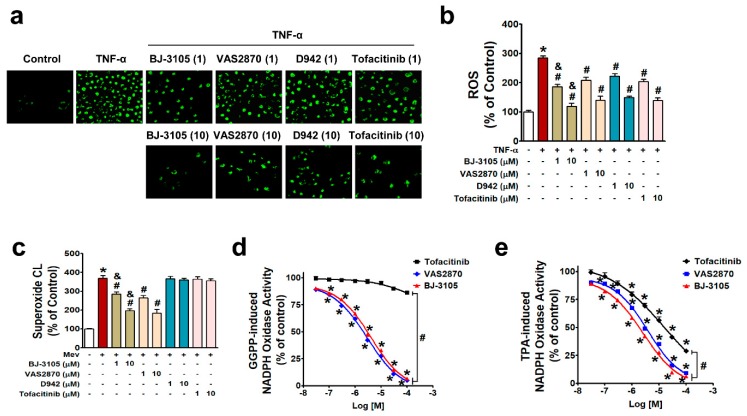
Inhibitory effects of BJ-3105, tofacitinib, D-942, and VAS2870 on TNF-α-, mevalonate, GGPP-, or TPA-induced ROS and NOX activities in HT-29 cells. (**a,b**) TNF-α-induced ROS levels in serum-starved HT-29 cells were measured by detecting DCF fluorescence using fluorescent microscopy (**a**) and microfluorometry (**b**). (**c**) Superoxide radical production was measured using a lucigenin chemiluminescence assay. (**d**,**e**) NOX activity was measured using lucigenin in cell extracts and GGPP (**d**) or TPA (**e**) as a stimulator. Results are presented as the means ± SEMs of three independent experiments. **p* < 0.05, versus vehicle-treated controls. ^#^*p* < 0.05, versus stimulator (TNF-α, mevalonate, GGPP, or TPA)-treated cells. ^&^*p* < 0.05, versus tofacitinib- or VAS2870-treated cells.

**Figure 6 ijms-21-03145-f006:**
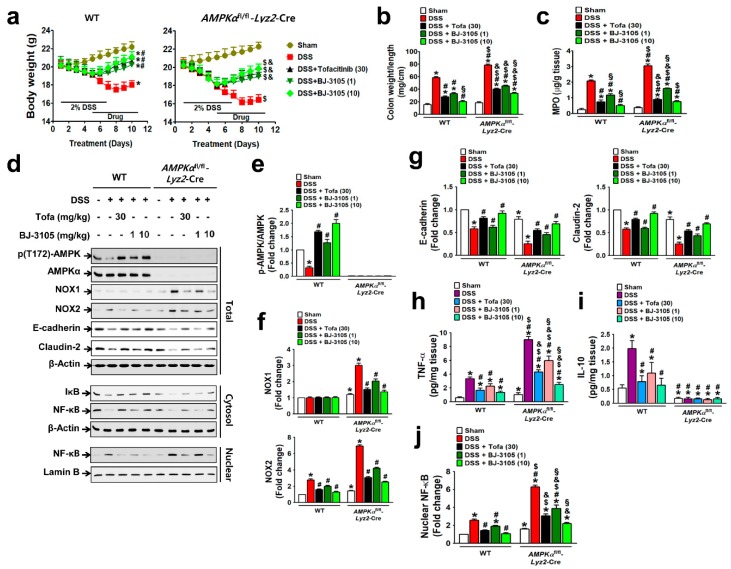
BJ-3105 was more effective than tofacitinib at ameliorating DSS-induced colitis in mice. (**a**–**c**) Inhibitory effects of tofacitinib and BJ-3105 on DSS-induced colitis in WT and *AMPKα*^fl/fl^-*Lyz2*-Cre mice were evaluated by measuring body weights (**a**), colon weight/unit length (**b**), and MPO level in colon tissues (**c**). (**d**–**g**) The effects of tofacitinib and BJ-3105 on gene expression levels in total protein, cytosolic, and nuclear extracts of colon tissues were determined by immunoblotting (**d**). Densitometry was used to assess p-AMPK/AMPK (**e**), NOX1/2 (**f**), and epithelial junction molecules, and E-cadherin and claudin-2 levels (**g**). (**h**,**i**) TNF-α (**h**) and IL-10 (**i**) levels in colon tissues were determined by ELISA. (**j**) Densitometry results of nuclear levels of NF-κB (**j**). **p* < 0.05, versus sham-operated WT controls. ^#^*p* < 0.05, versus DSS-treated WT mice. ^$^*p* < 0.05, versus sham-operated *AMPKα*^fl/fl^-*Lyz2*-Cre mice. ^&^*p* < 0.05, versus DSS-treated *AMPKα*^fl/fl^-*Lyz2*-Cre mice. ^§^*p* < 0.05, versus tofacitinib-treated WT or AMPKα KO mice.

**Figure 7 ijms-21-03145-f007:**
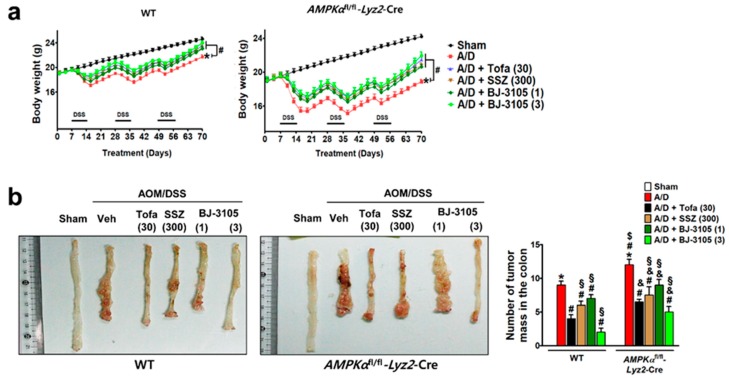
Efficacies of BJ-3105, tofacitinib, and sulfasalazine on tumor formation in AOM/DSS-treated mice. WT and *AMPKα*^fl/fl^-*Lyz2*-Cre mice were used for the AOM/DSS-induced colitis-associated cancer model experiment. WT and *AMPKα*^fl/fl^-*Lyz2*-Cre mice were used for the AOM/DSS-induced colitis-associated cancer model experiment. BJ-3105 (1 and 3 mg/kg), tofacitinib (30 mg/kg), or sulfasalazine (SSZ) (300 mg/kg) were administered orally. (**a**) Changes in body weight. (**b**) The representative view of the distal colon lumen. In the macroscopic examination, tumors larger than 2 mm in diameter were counted, and the numbers of tumors are presented as means ± standard errors (*n* = 5). **p* < 0.05, versus sham-operated WT mice. ^#^*p* < 0.05, versus AOM/DSS-treated WT mice. ^$^*p* < 0.05, versus sham-operated *AMPKα*^fl/fl^-*Lyz2*-Cre mice. ^&^*p* < 0.05, versus AOM/DSS-treated mice. ^§^*p* < 0.05, versus tofacitinib-treated WT or *AMPKα*^fl/fl^-*Lyz2*-Cre mice.

**Table 1 ijms-21-03145-t001:** Primer sequences used for genotyping.

Primer	Sequence 5′→3′
*Prkaa1* forward	CCCACCATCACTCCATCTCT
*Prkaa1* reverse	AGCCTGCTTGGCACACTTAT
*Lyz2*-Cre mutant	CCCAGAAATGCCAGATTACG
*Lyz2*-Cre common	CTTGGGCTGCCAGAATTTCTC
*Lyz2*-Cre wild type	TTACAGTCGGCCAGGCTGAC

Primers were obtained from Bioneer Corporation (Daejeon, South Korea).

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
