# Peer review of "Potent Inhibitory Effect of BJ-3105, a 6-Alkoxypyridin-3-ol Derivative, on Murine Colitis Is Mediated by Activating AMPK and Inhibiting NOX"

_ijms, 2020, doi:10.3390/ijms21093145_

Round 1
Reviewer 1 Report
As identified by the authors, targetting a high-level regulator of cellular metabolism like AMPK provides the opportunity to perhaps more successfully suppress inflammation than can be achieved by targetting specific pathways e.g. TNFα & JAK/STAT signalling. For this reason alone I think that the manuscript is of considerable scientific interest. I think that the manuscript provides sufficient evidence for BJ-3105 having a direct effect on AMPK activity which results in reduced pro-inflammatory signalling via ROS and cytokines more effectively than tofacitinib. My criticisms are pretty minor and overall the paper would fit for publication after a little revision.
INTRODUCTION
Biologic therapy has found favour as a part of a “top-down” approach to achieve remission in moderate to severe cases of UC, despite TNF therapy generally being less effective in UC. Although the basis for non-responsiveness is unclear, I think given the statement on line 61: “Nevertheless, anti-TNF-α therapy…” it should be pointed out that UC and CD are immunologically distinct disorders. While the first paragraph is fine, thereafter I thought that discussion should focus on UC alone considering the use of tofacitinib as comparator and the use of DSS-colitis as in vivo model.
METHODS
- Which colonic tissues were used to prepare homogenates, mucosa alone or deeper tissues included?
- No mention of IL-10 ELISA in methods, were these the same as used for the immunoblots?
- Instances throughout methods where exponents need to be denoted properly e.g. 1×10^5 or 1×10⁵ cells/mL.
RESULTS
Figure 2
- The immunoblot would suggest that there is an inhibition of phosphorylation of STAT-3 but the methods do not list which antibody was used or even which phosphorylation site was being detected. This could explain a reduction in intracellular IL-10 protein?
- What was the rationale behind the inclusion of JAK2 rather than JAK1 or JAK3? Aren’t JAK1/3 targets for the tofacitinib?
Figure 4
- Although HT-29 exists as a label in Fig 4a I’d recommend adding cell type to the legend in Fig 4.
Figure 6
- SSZ should be given as abbreviation of sulfasalazine.
- Counted tumour masses within the colon but it is unclear from the methods as to whether colonic tumours confirmed as adenoma histologically to distinguish from inflammatory lesions.
Author Response
Reviewer 1
As identified by the authors, .... My criticisms are pretty minor and overall the paper would fit for publication after a little revision.
INTRODUCTION
Biologic therapy has found favour as a part of a “top-down” approach to achieve remission in moderate to severe cases of UC, despite TNF therapy generally being less effective in UC. Although the basis for non-responsiveness is unclear, I think given the statement on line 61: “Nevertheless, anti-TNF-α therapy…” it should be pointed out that UC and CD are immunologically distinct disorders. While the first paragraph is fine, thereafter I thought that discussion should focus on UC alone considering the use of tofacitinib as comparator and the use of DSS-colitis as in vivo model.
--> Response: Thank you for valuable comments. We have pointed out that UC and CD are immunologically distinct disorders and anti-TNF-α therapy less effective in UC, by adding and correcting the sentence, such as “CD and UC are immunologically distinct IBD diseases. CD is primarily associated with T-helper type 1 (Th1) immune responses with elevated levels of IL-12, interferon (IFN)-γ, and TNF-α, whereas UC is described as a Th2-mediated disease in which IL-4, IL-5, and IL-13 levels are increased [11]. Although the underlying mechanism of differential response is unclear, anti-TNF therapy is generally less effective in UC. In many IBD patients who do not respond to anti-TNF therapy, other inflammatory pathways such as the interleukin (IL)-1β pathway were upregulated [12]”
Which colonic tissues were used to prepare homogenates, mucosa alone or deeper tissues included?
--> Response: We used entire colon wall, and this is now specified in the Methods part, as “MPO levels were measured in homogenates using the entire tissues of the colon wall.”
No mention of IL-10 ELISA in methods, were these the same as used for the immunoblots?
--> Response: Thank you for pointing the missing method. We have added ELISA method measuring TNFα and IL-10 in Methods part (Section 4.11).
Instances throughout methods where exponents need to be denoted properly e.g. 1×10^5 or 1×10⁵ cells/mL.
--> Response: Thank you for the corrections. Those are corrected into 1×10⁵, 1×108, 2×105, and 5×105.
Figure 2
The immunoblot would suggest that there is an inhibition of phosphorylation of STAT-3 but the methods do not list which antibody was used or even which phosphorylation site was being detected. This could explain a reduction in intracellular IL-10 protein?
--> Response: In the revised version, we have added the antibody information in the Materials section, such as “AMPK-α rabbit polyclonal antibody, rabbit monoclonal antibodies specific to p-AMPK-α (Thr172), JAK2, p-JAK2 (Tyr1008), and occludin, and mouse monoclonal antibodies detecting STAT3, p-STAT3 (Tyr705), cleaved caspase 1, and IL-1β were purchased from Cell Signaling Technology Inc. (Boston, MA, USA).” We also specified the phosphorylation site detected by each antibody in the Fig. 2.
Phosphorylation of Tyr1007 and Tyr1008 in the activation loop of the kinase domain plays an essential role in Jak2 activation, and phosphorylation of Tyr705 in STAT3 by JAK2 induces STAT3 activation, dimerization, and transloction to the nucleus.
IL-10 expression was upregulated by IL-6 (Fig. 2) and LPS (Fig. 3) in HT-29 cells and mouse macrophages, respectively. Of importance, IL-10 expression was up-regulated in colon tissues treated with DSS in WT mice, but IL-10 expression was completely suppressed in colon tissues of AMPKαfl/fl-Lyz2-Cre mice, regardless of DSS treatment (Fig. 6i). As we discussed in Discussion part (page 10, lines 301-303), such as “our current study also showed complete suppression of IL-10 expression in colon tissues of AMPK-deleted mice which still express JAK2 and STAT3 proteins, indicating that without AMPK, IL-10 production is not possible”, IL-10 decrease is not associated with JAK/STAT3.
What was the rationale behind the inclusion of JAK2 rather than JAK1 or JAK3? Aren’t JAK1/3 targets for the tofacitinib?
--> Response: JAK1/2/3 are the targets of tofacitinib. Regarding tissue distribution, JAK1 and JAK2 are ubiquitously expressed, whereas JAK3 is primarily confined in the hematopoietic system, particularly myeloid and lymphoid cells. In the present study of HT-29 colonic epithelial cells, we compared inhibitory effects of BJ-3105 and tofacitinib on IL-6-induced signaling, which is known to activate STAT3 via JAK1/2. In addition, it is also reported that TNF-α-stimulated IL-6 production is suppressed by inhibition of JAK2 (BioMed Research International. 2013;2013:8. doi: 10.1155/2013/580135.580135). Based on those reports, we checked JAK2 and STAT3 changes.
Figure 4
Although HT-29 exists as a label in Fig 4a I’d recommend adding cell type to the legend in Fig 4.
--> Response: According to reviewer’s comments, HT-29 is specified in the Fig. 4 legend.
Figure 6
SSZ should be given as abbreviation of sulfasalazine.
--> Response: SSZ was used in the experiments of Fig. 7. In the legend of Fig. 7, we specified SSZ, as “BJ-3105 (1 and 3 mg/kg), tofacitinib (30 mg/kg), or sulfasalazine (SSZ) (300 mg/kg) were administered orally.”
Counted tumour masses within the colon but it is unclear from the methods as to whether colonic tumours confirmed as adenoma histologically to distinguish from inflammatory lesions.
--> Response: We have many experiences of murine colitis experiment as in Fig. 6, and we observed DSS (2%)-induced inflammation caused mucosa damage and ablation. However, in AOM/DSS model, mice were administered with 1% DSS at 3rd exposure, and they were sacrificed 2 weeks after 3rd DSS exposure. There was no obvious inflammation in distal colon, but polyp-type tumors were developed. In the macroscopic examination, we were able to count tumors larger than 2 mm in diameter.
Reviewer 2 Report
The study is strong and well designed. It is supported by convincing results.
However a major point need to be addressed.
The in vitro experiments were performed in HT-29, a well-known colorectal adenocarcinoma.
It would be necessary to perfomr the experiments by using normal colon cells to prove the toxicity of the substance and the efficacy of the treatment.
Author Response
The in vitro experiments were performed in HT-29, a well-known colorectal adenocarcinoma. It would be necessary to perfomr the experiments by using normal colon cells to prove the toxicity of the substance and the efficacy of the treatment.
-->Response: We have compared toxicity of BJ-3105 and tofacitinib in CCD-841, a normal colon epithelial cell line. The results are added as Fig. 1e. BJ-3105 toxic profile was similar to tofacitinib, and their IC50 concentrations that inhibit cell viability by 50% were greater than 100 μM.
Previously, we had efficacy comparison between HT-29 cells and CCD-841 cells, and found there was no difference in cellular responses between them including cytokine-induced adhesion ability and molecular expressions (Free Radical Biology and Medicine, 2014, 69: 377-389).
Round 2
Reviewer 2 Report
The authors revised the manuscript by following reviewer's suggestion, which improved the quality of the study.